# A mathematical model for automatic differentiation in machine learning

**Jérôme Bolte**\*
Toulouse School of Economics
Univ. Toulouse
Toulouse, France

**Edouard Pauwels**
IRIT, CNRS
Univ. Toulouse
Toulouse, France

## Abstract

Automatic differentiation, as implemented today, does not have a simple mathematical model adapted to the needs of modern machine learning. In this work we articulate the relationships between differentiation of programs as implemented in practice and differentiation of nonsmooth functions. To this end we provide a simple class of functions, a nonsmooth calculus, and show how they apply to stochastic approximation methods. We also evidence the issue of artificial critical points created by algorithmic differentiation and show how usual methods avoid these points with probability one.

## 1 Introduction

Optimization algorithms based on backpropagation oracles, and more generally on automatic or algorithmic differentiation (AD) [41, 39], are one of the most widely used training tools for modern learning architectures [14, 32, 15, 18, 20, 3, 16]. They often rely on popular numerical implementations as TensorFlow or PyTorch [1, 36]. However, for nonsmooth, nonconvex losses, AD does not have a stable theory [23, 25, 26, 2, 30, 28, 29, 12], matching the actual practice. We wish to present a simple mathematical framework addressing this issue. Let us progressively explain our approach.

### 1.1 What is backpropagation?

**Algorithmic differentiation acts on programs not on functions:** To convey this fact we carry out a small experiment in TensorFlow [1] with the function $\mathrm{relu} \colon t \mapsto \max\{0, t\}$, see Appendix A.2 for implementation details. Algorithmic differentiation is displayed in Figure 1, in particular, we have $\mathrm{relu}'(0) = 0$. Consider the two functions

$$\mathrm{relu}_2 \colon t \mapsto \mathrm{relu}(-t) + t, \qquad \mathrm{relu}_3 \colon t \mapsto \frac{1}{2}(\mathrm{relu}(t) + \mathrm{relu}_2(t)).$$

As mathematical functions on $\mathbb{R}$ these are *equal to* $\mathrm{relu}$. However TensorFlow returns $\mathrm{relu}_2'(0) = 1$ and $\mathrm{relu}_3'(0) = 1/2$ (Figure 1). Indeed, AD does not act on functions, but on their representations, i.e., on programs. Different programs implementing the same function may provide different results, beyond numerical precision; we refer to this as the spurious behaviour of AD for nonsmooth functions[2]. Let us explore this phenomenon further. The function $\mathrm{zero} \colon t \mapsto \mathrm{relu}_2(t) - \mathrm{relu}(t)$, outputs constantly 0 but AD gives $\mathrm{zero}'(0) = 1$. More generally, one can modify the value of the derivative of a given function at prescribed arguments (Figure 1). This may generate artificial critical points; for instance $x \to x - \mathrm{zero}$ is the identity but its derivative at 0 according to AD is 0.

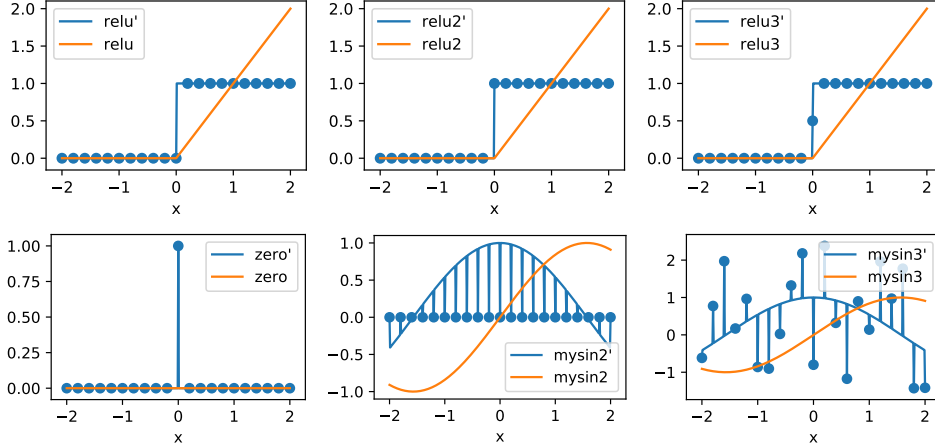

Figure 1: Top: AD applied to relu and two different implementations of the same function. Bottom: Algorithmic differentiation of a constant function, creation of artificial critical point or arbitrary derivatives at prescribed arguments for the sine function.

This discussion was limited to univariate functions, but these pathologies grow in size and in complexity when occurring in higher dimensions. Besides, as the "compositional depth" of functions increases the phenomenon gets more complex, making the geometry of artificial point difficult to grasp.

**Canonical surjection between programs functions:** Numerical programs combine basic mathematical functions within an algorithm and return an output. This can be understood in two ways:

- Computer science: it is a sequence of instructions with numerical inputs-outputs,
- Mathematics: the program is a function[3] of its arguments.

It is tempting to identify both, but functions can be represented by different programs. This defines a surjection $\mathcal{F}$ mapping a program to a function (in the class of functions "accessible through coding").

**Algorithmic differentiation:** As presented above, AD is an operation on programs, $\mathcal{A}$ which takes as argument a program and returns a program with the same input variables. This operation can be "pushed" to the space of functions using the canonical surjection $\mathcal{F}$. Remarkably, if we restrict ourselves to programs $\mathcal{P}$ which only smoothly combine smooth functions, then we have the following fundamental relation, depicted in Figure 2:

$$\mathcal{F}(\mathcal{A}(\mathcal{P})) = \nabla \mathcal{F}(\mathcal{P}). \tag{1}$$

In other words, algorithmic differentiation of a program which smoothly combines smooth functions, is equivalent, through the canonical surjection, to derivation.

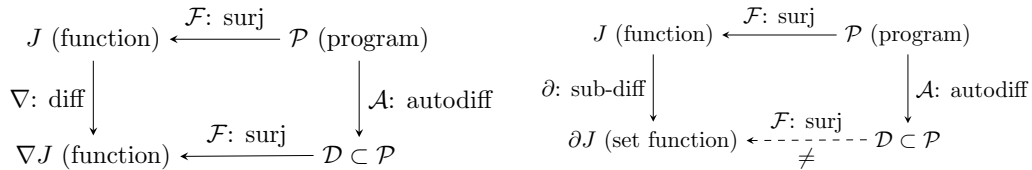

Figure 2: Left: Algorithmic differentiation applied to programs combining smooth functions in a smooth way, the diagram commutes. Right: Algorithmic differentiation in nonsmooth settings, connection with known notion of generalized derivative is much less clear.

However practitioners use AD and backpropagation beyond smooth programs with nonsmooth elementary functions or program branching for instance. Can we find a proper operational interpretation of this widespread practice?

**Algorithmic differentiation cannot be represented through a variational operator**  At first, it is tempting to simply use AD to induce a differential operator on functions generalizing classical differentiation. This operator, say $\partial^A$, should:

(a) encompass the outputs of algorithmic differentiation for all functions

(b) be such that 0 is an element of $\partial^A(\mathrm{relu})$ at 0.

Unfortunately such an operator does not exist:

**Theorem 1 (Algorithmic differentiation does not induce an operator on functions)** *There is no nontrivial operator on functions satisfying* $(a)$ *and* $(b)$.

## 1.2  Contribution and related work

We address this impossibility result and provide a class of functions together with an operational nonsmooth differential calculus which is able to cope with spurious behaviours.

**Elementary selections and selection derivatives:**  We introduce a new class of nonsmooth nonconvex functions, encompassing most objective functions met in machine learning, having appealing stability properties. This allows us to define simple differential objects called selection derivatives. Selection derivatives turn out to have an operational calculus adapted to the analysis of many learning methods, as backpropagation or stochastic first order methods. They thus provide an operational model to capture nonsmooth AD as implemented in current numerical software.

**Algorithmic differentiation, algorithms**  This framework allows to formalize properly the relationships between, functions, algorithmic differentiation and capture the corresponding notion of critical points as met in practice. These characterize the set of attractors (limit points) for stochastic approximation algorithms based on nonsmooth backpropagation [37, 4, 31, 5, 13]. It is important to stress that these attractors, which models sharply the whole scope of AD-induced stationnarity, are different from the traditional notions as Clarke criticality [17, 38, 20]. This is described in Theorems 3 and 4.

**Avoidance of traps:**  As sketched above and in the introduction AD produces artificial critical points, i.e. stationary points which are not Clarke critical. These points have a parasitic nature which could be detrimental to training purposes, were they met in practice. We show that randomly initialized mini-batch stochastic gradient method do not lead to artificial critical points (Theorem 4). This result applies to modern machine learning software libraries based on AD [1, 36], seen as performing operation over the reals, without any modification. Although AD may have unpredictable behavior in nonsmooth contexts, both theoretically and numerically, this result justifies theoretically that the practical impact is somewhat negligible in the context of common machine learning usage.

**Related work:**  Spurious behaviour of AD in nonsmooth context has been investigated in [23, 25, 26, 27, 2, 30, 12]. In particular, [27, 30] considers qualification conditions allowing to construct AD algorithms which compute proper Clarke subgradients [17, 38, 20]. However qualification is extremely hard to check and almost impossible to enforce in practice. Let us also mention [2] which uses the notion of lexicographic derivatives, but, at this day, algorithmic computations are limited to forward mode for the moment which is of little use in machine learning.

[23, 25, 27, 26, 28, 29] use settings closer to ours. Piecewise smooth functions, selection derivatives and their variational properties are extensively described in [40]. Our approach differs because we adopt more stringent definitions and rigidity assumptions, which allows in turn for much stronger properties. For instance, we fully treat backward algorithmic differentiation which is the most useful tool in machine learning.

Altogether, our contribution is an accessible and elementary framework for the conservative fields recently introduced in [12], without explicitly requiring the introduction of semialgebraic geometry and o-minimal structures [21, 19].

Stochastic approximation algorithms [37, 4, 31, 5, 13] are widely used in machine learning contexts [39, 14, 35, 32, 15, 18, 16]. For example [20] describes asymptotics of stochastic subgradient algorithms in nonsmooth, nonconvex settings. In contrast, we do not assume access to subgradients and instead explicitly model the behaviour of AD in optimization contexts. Our convergence results are based on [12], complemented by a new result on "the avoidance of critical traps" in the line of [7] in the context of long run convergence.

**Notations** The ambient space is Euclidean $\mathbb{R}^p$. For each $k$, $e_k$ is the $k$-th vector of the canonical basis. We use $D\colon \mathbb{R}^m \rightrightarrows \mathbb{R}^q$ for set valued functions, *i.e* functions from $\mathbb{R}^m$ to the subsets of $\mathbb{R}^q$. The convex hull of $A \subset \mathbb{R}^p$ is denoted by $\mathrm{conv}(A)$. All proofs are postponed to the Appendix.

## 2 Basic piecewise differentiable functions and selection gradient

We introduce a simple but vast class of functions that model rigorously the machine learning models and losses for applications such as deep learning.

**Definition 1 (Elementary (log-exp) functions)** *Elementary (log-exp) functions* are functions on $\mathbb{R}^p$ described by a finite compositional expression involving basic operations, $+, -, \times, /$ as well as affine mappings, exponential and logarithms, inside their domain of definition. We denote by $\mathcal{E}$ the set of elementary functions in any dimension $p$.

Examples include polynomials, logistic loss, boosting loss, Gaussian likelihood. Observe that the corresponding functions are $C^\infty$ smooth on their open domains. Note also that if log and exp are not present we obtain the field of rational functions. See Remark 1 in Appendix A.3.

**Definition 2 (Elementary index)** $s\colon \mathbb{R}^p \mapsto \{1, \ldots, m\}$ *is an elementary (log-exp) index if the set* $\{x \in \mathbb{R}^p,\, s(x) = i\}$ *is the solution set of a finite number of inequalities and equalities involving elementary functions on $\mathbb{R}^p$. The set of such functions is denoted by $\mathcal{I}$ (for any input dimensions $p$).*

**Examples:** The Heaviside function, the index of the largest or $k$-th largest element in a vector, the sign pattern of a vector in $\mathbb{R}^p$ which is indexed by integers from 1 to $2^p$.

**Definition 3 (Elementary selection)** Let $f\colon \mathbb{R}^p \mapsto \mathbb{R}$ be continuous. We say that $f$ has an *elementary (log-exp) selection* $(s, f_1, \ldots, f_m)$ if $s\colon \mathbb{R}^p \mapsto (1, \ldots, m)$ is an elementary index in $\mathcal{I}$ and for $i = 1 \ldots, m$, $f_i\colon \mathbb{R}^p \mapsto \mathbb{R}$ are elementary functions in $\mathcal{E}$, such that for all $x \in \mathbb{R}^p$,

$$f(x) = f_{s(x)}(x). \tag{2}$$

The $m+1$-uplet $(s, f_1, \ldots, f_m)$ is a *representation* of $f$, and $f$ admits an *elementary (log-exp) selection*. The class of such functions is denoted by $\mathcal{S}_{\log \exp}$ or simply here $\mathcal{S}$. This extends to functions from $\mathbb{R}^p$ to $\mathbb{R}^m$ by applying a coordinatewise definition with a common elementary index.

Observe that the representation is *never* unique, both in $s$ and in the sequence $f_1, \ldots, f_m$. The ReLU, hinge loss, maximal entry, $k$-th largest entry functions are elementary selections. Note also that continuity is part of the definition.

**Proposition 1 (Stability of $\mathcal{S}$ by $\circ, +, \times$)** *The class $\mathcal{S}$ of elementary selections is stable by composition, sum and product.*

The class $\mathcal{S}$ is close to the one of piecewise $C^k$ functions, see e.g [40], but it is also much more disciplined since indices and functions are required to satisfy strong "log-exp" rigidity assumptions.

### 2.1 Selection derivative

Functions in $\mathcal{S}$ can be associated with a flexible notion of generalized derivative based on the selection structure of the underlying function.

**Definition 4 (Selection gradient)** (i) Let $f \colon \mathbb{R}^p \mapsto \mathbb{R}$, in $\mathcal{S}$ with selection $(s, f_1, \ldots, f_m)$. We set the *selection derivative of $f$ with respect to $s$* to be

$$\widehat{\nabla}^s f \colon x \mapsto \nabla f_{s(x)}(x). \tag{3}$$

This extends to multivariate outputs by applying the definition coordinatewise, which leads to a notion of a *selection Jacobian* denoted by $\widehat{J}^s$.

(ii) Given a function $f \in \mathcal{S}$, a *selection derivative* is a derivative of the form (3) for a given representation. In that case a selection derivative of $f$ is merely denoted by $\widehat{\nabla} f$.

**Example:** Set for all $x \in \mathbb{R}$, $f_1(x) = 0$, $f_2(x) = x$ and $s(x) = 1$ for $x \leqslant 0$ and $s(x) = 2$ for $x > 0$. This this defines the relu function and its selection derivative at $0$ is $0$. See more in Appendix A.2.

**Remark:** (a) $\widehat{\nabla} f$ is different from any known notion of subgradient. Set for all $x \in \mathbb{R}$, $f_1(x) = 0$, $f_2(x) = x$ and $s(x) = 1$ for $x \neq 0$ and $s(0) = 2$. This defines a elementary selection for the null function however, $\widehat{\nabla}^s f(0) = 1$. This is the zero function of the introduction.

(b) This formalizes what one would obtained by differentiating a code with all decision branches frozen and hence represents the numerical output of AD (see 4). Note that one only needs one branch and do not need to explore all possible outcomes, avoiding combinatorial explosion.

The properties of selection derivatives might seem too liberal at first sight and too disconnected from the original function, but this is not the case as shown below.

**Proposition 2 (Integration along segments)** *Let $f \colon \mathbb{R}^p \mapsto \mathbb{R}$ be in $\mathcal{S}$, with elementary selection $(s, f_1, \ldots, f_m)$. Then $f$ is locally Lipschitz and for all $y, x$ in $\mathbb{R}^p$.:*

$$f(y) - f(x) = \int_0^1 \left\langle y - x, \widehat{\nabla}^s f(x + t(y - x)) \right\rangle dt$$

**Proposition 3 (Gradient almost everywhere)** *Let $f \colon \mathbb{R}^p \mapsto \mathbb{R}$ be in $\mathcal{S}$, with elementary selection $(s, f_1, \ldots, f_m)$. There exists sets $U_1, \ldots, U_N$ with nonempty interior such that $\bigcup_{i=1}^N \mathrm{cl}(U_i) = \mathbb{R}^p$ and for each $i = 1$, and for all $x$ in the interior of $U_i$, $\widehat{\nabla}^s f(x) = \nabla f(x)$. Furthermore, the $U_i$ are solution sets of equations and inequalities involving functions in $\mathcal{E}$.*

**Remark:** Although less transparent, Proposition 2 is not a consequence of Proposition 3. Both results crucially rely on the rigidity of elementary functions in $\mathcal{E}$ (Definition 3), not only on their piecewise smoothness. This a central novelty of our approach.

## 2.2 A calculus for selection derivatives

One has an unusual differential calculus: although it does not involve the linearity of some (sub)differential operator, the selection derivative of a sum gives a sum of selection derivatives provided that the selection is refined.

**Proposition 4 (Chain rule)** *Let $F \colon \mathbb{R}^{p_1} \mapsto \mathbb{R}^{p_2}$ such that each of its coordinate $f_i$, $i = 1 \ldots p_2$, is in $\mathcal{S}$ and $g \colon \mathbb{R}^{p_2} \mapsto \mathbb{R}$, $g \in \mathcal{S}$. Consider a selection Jacobian for $F$, $\widehat{J}_F \colon \mathbb{R}^{p_1} \mapsto \mathbb{R}^{p_2 \times p_1}$*

$$x \mapsto \begin{pmatrix} \widehat{\nabla} f_1(x)^T \\ \vdots \\ \widehat{\nabla} f_q(x)^T \end{pmatrix} \tag{4}$$

*Then $g \circ F \in \mathcal{S}$ and the function $x \mapsto \widehat{J}_F(x)^T \widehat{\nabla} g(F(x))$ is a selection derivative for $g \circ F$.*

Proposition 4 extends readily to the case when the outer function $g$ is multivariate. For example, we have a sum rule $\widehat{\nabla}(f + g) = \widehat{\nabla} f + \widehat{\nabla} g$ for full-domain functions $f, g$ in $\mathcal{S}$. Indeed, if $F_1$ and $F_2$ are elementary selections then $F_1 \circ F_2 \in \mathcal{S}$ and

$$\widehat{J}_{F_1 \circ F_2} = (\widehat{J}_{F_1} \circ F_2) \times \widehat{J}_{F_2}. \tag{5}$$

# 3 Programs and elementary selections

Numerical programs encode numerical functions by combining elementary functions using a predecessor relation which models program execution. In what follows, $m$ can be seen as an estimate of the memory footprint of a program[4], while $p$ and $q$ the number of inputs and outputs respectively.

Given positive integers $m \geqslant p + q$, a *predecessor relation* is a set valued map $\mathtt{pr}\colon \{1, \ldots, m\} \rightrightarrows \{1, \ldots, m\}$ such that

- For $i \in \{1, \ldots, m\}$ and $j \in \mathtt{pr}(i)$, $j < i$.   • For $i \in \{p + 1, \ldots, m\}$, $\mathtt{pr}(i)$ is nonempty.

A predecessor relation induces a partial order on the set of integers from 1 to $m$ and hence can be represented by a directed acyclic graph [34, Theorem 9.4.9]. Given $(p, q, m)$ and a predecessor relation $\mathtt{pr}$, a elementary function sequence $\mathcal{G} = (g_i)_{i=p+1}^m$ is a set of functions such that $g_i \colon \mathbb{R}^{|\mathtt{pr}(i)|} \mapsto \mathbb{R}$, and $g_i \in \mathcal{S}$, for all $i = p + 1, \ldots, m$. A program $P$ is then given by the data $P = (p, q, m, \mathtt{pr}, \mathcal{G})$, while its *evaluation* is described in Algorithm 1. We denote by $\mathcal{P}$ the set of programs, and $\mathcal{P}_{p,q}$ when input-output dimensions have to be made explicit.

By definition a program encodes a function, but the representation is not unique. We express this fact below through the canonical surjection $\mathcal{F}$ of the introduction.

---

**Algorithm 1:** Program evaluation

---

**Program data:** $p, q \geqslant 1$, $m \geqslant p + q$, $\mathtt{pr}$ a predecessor relation, $\mathcal{G} = (g_i)_{i=p+1}^m$ an adapted function sequence.

**Input:** $x = (x_1, \ldots x_p)$
1: **for** $k = p + 1, p + 2, \ldots m$ **do**
2:    $x_k = g_k(x_{\mathtt{pr}(k)})$ where
      $x_{\mathtt{pr}(k)} = (x_i)_{i \in \mathtt{pr}(k)}$.
3: **end for**
**Return:** $y := (x_j)_{j=m-q+1}^m$.

---

The following proposition illustrates the fact that practitioners *implicitly* implement selection functions when writing programs.

**Proposition 5 (Programs represents elementary selections)** *Through its input-output correspondence each program $P$ of the form* (3) *induces a function which is an elementary selection. In other words $\mathcal{F}(P) \in \mathcal{S}$.*

# 4 Algorithmic differentiation and a variational model

Algorithmic differentiation is based on the idea of propagating infinitesimal variations in a program $P$ through the chain rule, either forward or backward.

---

**Algorithm 2:** Algorithmic differentiation computes selection gradients

---

**Program data:** $p \geqslant 1$, $m \geqslant p + 1$, $\mathtt{pr}$ a predecessor relation, $\mathcal{G} = (g_i)_{i=p+1}^m$ an adapted function sequence.

**Input:** variables $(x_1, \ldots x_m)$ computed by Algorithm 1, $d_i = (d_i[j])_{j=1}^{|\mathtt{pr}(i)|} = \widehat{\nabla} g_i(x_{\mathtt{pr}(i)})$, $i = p + 1 \ldots m$

1: **Forward mode:**
2: Initialize: $\frac{\partial x_k}{\partial x} = e_k$,
   $k = 1, \ldots, p$.
3: **for** $k = p + 1, \ldots m$ **do**
4:
$$\frac{\partial x_k}{\partial x} = \sum_{j \in \mathtt{pr}(k)} \frac{\partial x_j}{\partial x} d_k[j]$$
   where $x = (x_1, \ldots, x_p)$.
5: **end for**
**Return:** $\frac{\partial x_m}{\partial x}$.

1: **Backward mode:**
2: Initialize: $v = e_m$
3: **for** $t = m, \ldots p + 1$ **do**
4:    **for** $j \in \mathtt{pr}(t)$ **do**
5:       Update coordinate $j$ of $v$:
$$v[j] := v[j] + v[t]d_t[j]$$
6:    **end for**
7: **end for**
**Return:** $(v[1], v[2], \ldots, v[p])$.

Consider Algorithm 1, and assume for simplicity that $q = 1$. The program can be seen as the implementation of $m - p$ successive transformations on $\mathbb{R}^m$, of the form

$$G_k \colon \mathbb{R}^m \mapsto \mathbb{R}^m$$
$$x \mapsto x + e_k(g_k(x_{\mathbf{pr}(k)}) - x_k),$$

for $k = p + 1, \ldots, m$ which belong to $\mathcal{S}$. Algorithm 2 combines gradients dynamically along two modes: forward or backward. Let us describes these two forms.

Fix $x \in \mathbb{R}^m$. After applying Algorithm 1, for each $k$, let $d_k \in \mathbb{R}^m$ be the selection gradient $\widehat{\nabla} g_k(x_{\mathbf{pr}(k)})$, appending 0 to non dependant coordinates. A selection Jacobian of $G_k$ (at $x$) is given by

$$\widehat{J}_{G_k} = I - e_k e_k^T + e_k d_k^T$$

Denote by $J_p \in \mathbb{R}^{m \times p}$, the matrix whose entries are 0, except for diagonal entries which are 1. In Algorithm 2, the forward mode computes

$$e_m^T \widehat{J}_{G_m} \ldots \widehat{J}_{G_{p+1}} J_p = e_m^T \left( I - e_m e_m^T + e_m d_m^T \right) \ldots \left( I - e_{p+1} e_{p+1}^T + e_{p+1} d_{p+1}^T \right) J_p$$

which is a selection Jacobian thanks to the chain rule in (5). On the other hand the backward mode computes

$$J_p^T \left( I + d_{p+1} e_{p+1}^T \right) \ldots \left( I + d_m e_m^T \right) e_m.$$

This quantity turns out to be the same as the one computed by the forward mode thanks to:

**Lemma 1** *Let $p, m \in \mathbb{N}$, $0 < p < m$. Assume that for $i = p + 1, \ldots, m$ we have $d_i \in \mathbb{R}^m$. Then we have*

$$P_p \left( I - e_{p+1} e_{p+1}^T + d_{p+1} e_{p+1}^T \right) \ldots \left( I - e_m e_m^T + d_m e_m^T \right) = P_p \left( I + d_{p+1} e_{p+1}^T \right) \ldots \left( I + d_m e_m^T \right)$$
(6)

*where $I \in \mathbb{R}^{m \times m}$ is the identity matrix and $P_p \in \mathbb{R}^{m \times m}$ denotes the projection on the first $p$ coordinates.*

Denote by $\mathcal{A} : \mathcal{P}_{p,1} \to \mathcal{P}_{p,p}$ the algorithmic-differentiation operator. This establishes the following fundamental fact which is at the root of this work. This result asserts that practitioners *implicitly* implement selection derivatives when writing numerical programs and calling forward or backward AD on these programs.

**Theorem 2 (Algorithmic differentiation outputs a selection gradient)** *Algorithmic differentiation of a given program, i.e., $\mathcal{A}(P)$, outputs a selection derivative of the underlying numerical function. In other words there exists a representation of the numerical function $\mathcal{F}(P)$ with elementary index $s$ such that:*

$$\mathcal{F}(\mathcal{A}(P)) = \widehat{\nabla}^s \mathcal{F}(P).$$

## 5 Algorithmic differentiation at work

### 5.1 Selection derivatives, conservative fields and Clarke subgradient

The asymptotic study of first-order optimization methods implies limiting processes and necessitates thus the introduction of graph closed operators. Given a representation for $f$, we may construct such a convex-valued mapping pointwise as follows[5].

**Definition 5 (Representation minimal operator)** Let $f \in \mathcal{S}$ with elementary selection $(s, f_1, \ldots, f_m)$. For any $x \in \mathbb{R}^p$, set $I(x) = \{i \in \{1, \ldots, m\}, \, f(x) = f_i(x)\}$. The index closure of $\widehat{\nabla}^s f$ is given by the set valued map

$$D_f^s \colon \mathbb{R}^p \rightrightarrows \mathbb{R}^p$$
$$x \rightrightarrows \mathrm{conv}\left(\{\nabla f_i(x), \, i \in I(x)\}\right).$$

where the double arrows express that the map has values in subsets of $\mathbb{R}^p$, much like subgradients, and conv denotes the convex hull.

The role of $D_f^s$ is to capture all possible outputs of AD including all possible program branches. Of course, due to combinatorial explosion, this quantity is intractable in practice. Its introduction here is only instrumental, we do not use it in algorithms, we just need to access one of its element, for example using a selection derivatives, obtained from AD. A point $x$ satisfying $0 \in D_f^s(x)$ is called a *selection critical point*. We will often drop the index $s$ and write $D_f = D_f^s$.

The two following results highlight crucial properties of $D_f$ in terms of optimization, they again rely on the rigidity constraint of elementary functions.

**Theorem 3** *Let $f \in \mathcal{S}$ with elementary selection $(s, f_1, \ldots, f_m)$ and $D_f$ be as in Definition 5. Then $D_f$ is conservative for $f$, that is for all absolutely continuous curves $\gamma \colon [0,1] \mapsto \mathbb{R}^p$, for almost all $t \in [0,1]$, $f \circ \gamma$ is differentiable and*

$$\frac{d}{dt} f(\gamma(t)) = \langle v, \dot{\gamma}(t) \rangle, \qquad \forall v \in D_f(\gamma(t)).$$

The previous result generalizes Proposition 2 by allowing to integrate arbitrary selections along absolutely continuous curves. This connects our work to the general setting of [12], note that $D_f$ has a closed graph thanks to Proposition 6 in Appendix A.3.

In [40], the author considers the *essential index set*, for each $x \in \mathbb{R}^p$,

$$S_E(x) = \{i \in \{1, \ldots, m\}, \, x \in \mathrm{cl}(\mathrm{int}(\{y, \, f(y) = f_i(y)\}))\} \subset S(x).$$

Considering Definition 5 with $S_E(x)$ instead of $I(x)$ leads to the Clarke subgradient, which can also be defined as

$$\partial^c f(x) = \mathrm{conv}\{d \in \mathbb{R}^p : \exists x_k \in \Delta_f, x_k \to x, \nabla f(x_k) \to d\}$$

where $\Delta_f$ is the dense set of differentiability points of $f$. While $I(x)$ can be computed pointwise (check finitely many equalities), it might be very hard to check membership in $S_E(x)$ without restrictive qualification conditions on programs [30].

**Illustration with ReLU and sorting:** (a) Set for all $x \in \mathbb{R}$, $f_1(x) = 0$, $f_2(x) = x$, $s(x) = 1$ for $x \leqslant 0$ and $s(x) = 2$ for $x > 0$. This is relu. In this case $D_f = \partial\mathrm{relu}$, the convex subgradient.
(b) Let $F \colon \mathbb{R}^p \mapsto \mathbb{R}^p$ to be the sorting function which associates to $x$ a vector $Px$ where $P$ is any permutation such that $Px$ belongs to the set of vectors which values are sorted in descending order coordinatewise. $F$ obviously has an elementary selection and the construction which we have proposed leads to

$$D_F \colon x \mapsto \mathrm{conv} \{P \in \Delta, \quad Px = F(x)\},$$

where $\Delta$ denotes the set of permutation matrices of size $p \times p$. Then $D$ is a conservative mapping for $F$ and it actually corresponds to the Clarke Jacobian.

## 5.2 Convergence of gradient type algorithm and criticality of limit points

Optimization processes in learning are supposed to provide at least a critical point $x$ of the loss, i.e. a point satisfying $0 \in \partial^c f(x)$. When using AD one enlarges the definition of criticality into $0 \in D_f(x)$ and *artificial critical points* appear, they satisfy $0 \notin \partial^c f(x)$ and $0 \in D_f(x)$. Artificial critical points could possibly trap the optimization process in strongly non-optimal situations, we thus have to determine if they have an impact on learning phases.

We consider the problem

$$\min_{x \in \mathbb{R}^p} J(x) = \frac{1}{n} \sum_{i=1}^{n} f_i(x) \tag{7}$$

where $f_i \colon \mathbb{R}^p \mapsto \mathbb{R}$, $f_i \in \mathcal{S}$, $i = 1, \ldots, n$. We consider the following algorithm, given $x_0 \in \mathbb{R}^p$, a sequence of positive step sizes $(\gamma_k)_{k \in \mathbb{N}}$ and a sequence of *iid* indices $(I_k)_{k \in \mathbb{N}}$ taken uniformly in the nonempty subsets of $\{0, \ldots, n\}$,

$$x_{k+1} = x_k - \gamma_k \widehat{\nabla} f_{I_k}(x_k) \text{ where } f_I = \frac{1}{|I|} \sum_{i \in I} f_i, \, I \subset \{1, \ldots, n\}. \tag{8}$$

Note that as discussed in Section 4 selection derivatives can be computed by AD if $f_i$ are given by the data of numerical programs as in (3), and could be far from usual notions of subgradients. Hence this algorithm models explicitly the training of a nonsmooth deep network using existing backpropagation implementations. Note that $J \in \mathcal{S}$ and that $1/n \sum_{i=1}^{n} \widehat{\nabla} f_i$ is a selection gradient for $J$ as stated in Proposition 4, denote by $\widehat{\nabla} J$ this quantity and $D_J$ the corresponding set valued field (Definition 5). The following result illustrates that selection critical points are the only attractors for the recursion and that generically such attractors are actually Clarke critical. The first result stands on the theory developed in [5]. The second parallels developments in [7] in the context of long run convergence. The spurious behaviour illustrated in Figure 1 does not affect asymptotics, for typical initialization.

**Theorem 4 (Convergence and insignificance of artefacts)** *Let for all $k$, $\gamma_k = c\alpha_k$ where $c \in (0,1]$ and $\alpha_k = o(1/\log k)$ and $K \subset \mathbb{R}^p$ be open. Assume that for all $c \in (0,1]$ and all $x_0 \in K$ the sequence in (8) is bounded almost surely.*

- *For all $x_0 \in K$, almost surely, $J(x_k)$ converges as $k$ tends to infinity and all accumulation points, $\bar{x}$, of $(x_k)_{k \in \mathbb{N}}$ are selection critical points: $0 \in D_J(\bar{x})$.*

- *For almost all $c \in (0,1]$, almost all $x_0 \in K$, and almost surely, any accumulation point, $\bar{x}$, of $(x_k)_{k \in \mathbb{N}}$ is Clarke critical: $0 \in \partial^c J(\bar{x})$.*

## 6  Conclusion

The current work departs from existing approaches to nonsmooth algorithmic differentiation in a fundamental way. We propose to study the backward mode of AD, as implemented in machine learning, without any modification. Our theoretical results model thus AD "as is", and our focus is precisely on its unpredictable behavior in a nonsmooth context, addressing an issue which is ubiquitous in machine learning. Our main contribution was to prove that, in a stochastic optimization context, this spurious behavior is essentially harmless from a theoretical point of view, providing justifications for the use of AD outside of its original domain of validity in machine learning.

We achieve our goal by modeling sharply common machine learning functions and their differentiation using selection derivatives, a known concept, which models the way AD differentiates nonsmooth programs. We restrict it to certain classes of elementary functions, opening the possibility to use powerful geometric techniques.

Further questions include convergence rates and complexity issues, hardly tackled at this day, let us mention the attempt of [43]. Our theory is limited to continuous functions and an interesting venue is to extend it to discontinuous functions, in view of treating ranking operations [10] ubiquitous in recommendation systems, or more generally differentiating through an argmax [6].

### Broader impact

One of the goals of the paper is to raise awareness about an important issue of in the training of ML methods: the spuriousness of AD. To address adequately this issue, we think it is necessary to include algorithmic differentiation explicitly in the study of optimization algorithms, a point of view which is largely ignored by today's machine learning community.

### Acknowledgments and Disclosure of Funding

The authors acknowledge the support of ANR-3IA Artificial and Natural Intelligence Toulouse Institute, Air Force Office of Scientific Research, Air Force Material Command, USAF, under grant numbers FA9550-19-1-7026, FA9550-18-1-0226, and ANR MasDol. J. Bolte acknowledges the support of ANR Chess, grant ANR-17-EURE-0010 and ANR OMS. The authors would like to thank anonymous referees for careful reading of this work and useful suggestions. The authors would like to thank N. Asher and S. Gerchinovitz for useful discussions. We also warmly thank J. Malick who triggered this research.

## Footnotes

\* Authors in alphabetical order.

[2] The validity domain of AD is restricted in theory to smooth functions [23], yet it is common practice to use it for nonsmooth functions.

[3]In the usual mathematical sense.

[4]We consider programs which do not overwrite values in memory

[5]Minimality relates to the *representation of the function*, not the function itself. This is the minimal convex-valued operator, constructed pointwise and guaranteed to be graph-closed.

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
