[Supplementary Material]

This is the appendix for "A mathematical model for automatic differentiation in machine learning".

# A   A more comprehensive discussion and auxiliary results

## A.1   Related work and contribution

The use of backward mode of algorithmic differentiation (AD) for neural network training expanded in the 80's, the most cited reference being [39]. However the theory applies to much more optimization problems, see for example [24]. Indeed, numerical libraries implementing the backward mode of AD were already available in the 90's for `FORTRAN` code [8, 9] or `C/C++` code [22], 30 years before the emergence of `python` libraries. These early implementation could differentiate virtually any code, but their domain of validity, i.e., the setting for which one could predict what the output would be, was restricted to differentiable functions evaluated on their (open) domain of differentiability.

This was well known to the AD community, see for example [23], and exploring further the domain of validity of AD, beyond mere differentiability, was already a vivid problem.

Let us mention [23] who used notions such as finite selection, "isolated criticalities", stable domain or regular arcs, and argued that "functions given by evaluation procedures are almost everywhere real analytic or stably undefined" where "undefined" meant that a nonsmooth elementary function is used in the evaluation process. For piecewise smooth functions which nonsmoothness can be described using the absolute value function (abs-normal form), [25] developed a piecewis linearisation formalism and local approximation related to AD, [26] proposed an AD based bundle type method. These developments are based on the notion of piecewise smooth functions [40] which we use in this work. More recently, [28] applied these techniques to single layer neural network training and [29] proposed to avoid the usage of subgradient "oracles" in nonsmooth analysis as they are not available in practice. In a similar vein, let us mention [2] study lexicographic derivatives, a notion of directional derivatives which satisfy a chain rule making them compatible by forward mode AD, and [43] who use directional derivatives in the context of local sampling stochastic approximation algorithms for machine learning.

Constraint qualification is known in nonsmooth analysis to ensure favorable behavior of chain rules of differential calculus for nonsmooth objects (see [38]). These already appeared in the context of piecewise smooth functions of Scholtes with the notion of "essential selections". Such an approach was used in [30] to propose an AD algorithm for subgradient computation under constraint qualification. Similarly [27] study first and second order optimality, in relation to AD using constraint qualification.

The current work departs from all these approaches in a fundamental way. We propose to study backward mode of AD, as implemented for nonsmooth functions by standard software (e.g. TensorFlow, PyTorch), without any modification, addition of operations or hypotheses. Our theoretical results model AD as implemented in current machine learning libraries. Contrary to previous works, our focus is precisely on the unpredictable behavior of AD in nonsmooth context. Our main contribution is to show that in a stochastic optimization context, this spurious behavior is essentially harmless from a theoretical point of view, providing justifications for the use of AD outside of its original domain of validity in machine learning.

At the time this paper was accepted, we learnt about a paper proposing an analysis close to ours [33]. The authors show that AD applied to programs involving piecewise analytic continuous functions, under analytic partitions, compute gradients almost everywhere. This is the counterpart of Proposition 3, replacing log-exp elementary function in Definitions 1 and 2, by analytic functions.

## A.2   Implementation of relu

The implementation of the $relu$ function used in Figure 1 is given by the function `tf.nn.relu` in Tensorflow software library [1]. This implementation corresponds to the selection function described in Section 2 and the same result may be obtained by an explicit implementation of this branching selection as illustrated in the following figure

```python
def myRelu(x):
    if x<=0:
        return 0
    else:
        return x

def myReluVec(x):
    return tf.map_fn(myRelu,x)
```

One can imagine an equivalent implementation of $relu$ with a slightly different branching involving a strict inequality, that would correspond to an equivalent implementation of the same function, but the computed derivative at $0$ is different due to the implementation

```python
def myReluBis(x):
    if x<0:
        return 0
    else:
        return x

def myReluBisVec(x):
    return tf.map_fn(myReluBis,x)
```

### A.3 Auxiliary results and remarks

**Remark 1 (Elementary piecewise differentiable functions)**
(a) The building blocks in the construction of $\mathcal{S}$ in Definition 3 could be modified and adapted to other needs. Besides, the results we present in this article would remain true if we added real analytic functions restricted to compact sets.
(b) Note also that in Definition 3, functions are actually real analytic on their (open) domain of definition. Yet their extension might not be analytic, as for instance the function $f : x \neq 0 \to \exp(-1/x^2)$ extended by $f(0) = 0$.
(c) The construction of elementary piecewise functions in Definition 3, does not coincide in general with some natural minimal o-minimal, but are contained in a larger such structure. For instance, when the basic bricks are polynomial functions, we obtain the field of rational functions which differs from the set of semi-algebraic functions.

**Proposition 6 ($D_f$ has a closed graph)** *As $k \to \infty$, assume that $x_k \to \bar{x} \in \mathbb{R}^p$ and $v_k \in D_f(x_k)$, $v_k \to \bar{v}$. Then $\bar{v} \in D(\bar{x})$.*

## B    Proofs

**Proof of Theorem 1:**   Recall the operator is denoted by $\partial^A$. Fix a function $f$, by point (a), the operator $\partial^A f$ should contain

$$\begin{cases} \mathbb{R}^p \rightrightarrows & \mathbb{R}^p \\ x \to & \{\mathcal{A}(P)(x) : \mathcal{F}(P) = f, P \in \mathcal{P}\} \end{cases}$$

Let us show that the graph of the above is $\mathbb{R}^p \times \mathbb{R}^p$. Assume $p = 1$ for simplicity. For real numbers $r, s$, consider the functions $f_{r,s} = f + r \operatorname{zero}(\cdot - s)$ which coincide with $f$ but whose form induces programs $P_{r,s}$ of $f$. These satisfy $\mathcal{F}(P_{r,s}) = f$ and $\mathcal{A}(P_{r,s})(s) \ni \mathcal{A}(f)(s) + r$. Since $r$ is arbitrary, $\partial^A f(s) = \mathbb{R}^p$ and since $s$ is arbitrary, we actually have

$$\operatorname{graph} \partial^A f = \mathbb{R}^p \times \mathbb{R}^p.$$

Since $f$ is arbitrary, we have shown that $\partial^A$ is trivial. $\qquad\square$

**Proof of Proposition 2:**   The proposition is a consequence of Theorem 3 and (11) but it admits a more elementary proof which we detail here. Fix $x, y \in \mathbb{R}^p$. Let us admit the following claim –whose independent proof is given in Section C.

**Claim 1** *There exists a finite set of numbers $0 = a_0 < a_1 < \ldots < a_N = 1$, such that for all $i \in 0, \ldots N - 1$, the function $t \mapsto s(x + t(y - x))$ is constant.*

Fix $i \in 0 \ldots, N-1$, and $j \in 1 \ldots m$ such that $f = f_j$ on $(x + a_i(y - x), x + a_{i+1}(y - x))$. Since $f_j \in \mathcal{E}_p$, it is $C^1$ and we have by the fundamental theorem of integral calculus

$$
\begin{aligned}
f(x + a_{i+1}(y - x)) - f(x + a_i(y - x)) &= \int_{a_i}^{a_{i+1}} \langle \nabla f_j(x + t(y - x)), y - x \rangle \, dt \\
&= \int_{a_i}^{a_{i+1}} \left\langle \widehat{\nabla} f(x + t(y - x)), y - x \right\rangle dt.
\end{aligned}
$$

The conclusion follows because

$$
\begin{aligned}
f(y) - f(x) &= \sum_{i=0}^{N-1} f(x + a_{i+1}(y - x)) - f(x + a_i(y - x)) \\
&= \sum_{i=0}^{N-1} \int_{a_i}^{a_{i+1}} \left\langle \widehat{\nabla} f(x + t(y - x)), y - x \right\rangle dt \\
&= \int_0^1 \left\langle \widehat{\nabla} f(x + t(y - x)), y - x \right\rangle dt.
\end{aligned}
$$

$\square$

**Proof of Proposition 3:** Constructs the sets $U_i$ by considering sets $V_j = \{x \in \mathbb{R}^p, \, s(x) = j\}$, $j = 1 \ldots m$, the proof of the following claim is postponed to Section C.

**Claim 2** *The boundary of each $V_j$ has zero measure and* $\mathrm{cl}\left( \cup_{i=j}^m \mathrm{int}(V_j) \right) = \mathbb{R}^p$.

Hence, we may define $U_1, \ldots, U_N$ by keeping only those sets with nonempty interior and take their closure. On each set $U_i$, $f$ is identical to $f_k$ for some $k$ and the result follows. $\square$

**Lemma 2** *Let $t \in \mathcal{I}$ be an elementary index on $\mathbb{R}^{p_2}$ and $F \colon \mathbb{R}^{p_1} \mapsto \mathbb{R}^{p_2}$ with each coordinate in $\mathcal{E}$, then $t \circ F$ is an elementary index on $\mathbb{R}^{p_1}$.*

**Proof :** Fix an arbitrary integer $i$ in the image of $t$, by Definition 2, there exists elementary functions $h_1, \ldots, h_J$, $J \in \mathbb{N}$ on $\mathbb{R}^{p_2}$ such that $t(y) = i$ if and only if $y \in K_i := \{z \in \mathbb{R}^{p_2}, \, h_j(z) \diamond_j 0, \, j = 1, \ldots J\}$ where $\diamond_j$ is an equality or inequality sign depending on $j$. Then $t(F(x)) = i$ if and only if $F(x) \in K_i$ which is equivalent to say that $x \in \tilde{K}_i := \{x \in \mathbb{R}^{p_1}, h_j(F(x)) \diamond_j 0, \, j = 1, \ldots J\}$. By Definition 1, $h_j \circ F$ is an elementary function for $j = 1, \ldots, J$ and $i$ was an arbitrary integer, this shows that we have an elementary index. $\square$

**Proof of Proposition 1:** Let $F \colon \mathbb{R}^{p_1} \mapsto \mathbb{R}^{p_2}$ such that each of its coordinate $f_i$, $i = 1 \ldots p_2$, is in $\mathcal{S}$ and $g \colon \mathbb{R}^{p_2} \mapsto \mathbb{R}$, $g \in \mathcal{S}$. We establish that $g \circ F$ is an elementary selection, the other cases are similar. We may consider all possible intersections of constant index domains across all coordinates of $F$ in $\{1, \ldots, p_2\}$. We obtain $(s, F_1, \ldots, F_m)$, an elementary selection for $F$ (each $F_i \colon \mathbb{R}^{p_1} \mapsto \mathbb{R}^{p_2}$ has coordinates in $\mathcal{E}$) . Consider $g \in \mathcal{S}$ with elementary selection $(t, g_1, \ldots, g_l)$. The composition $g \circ F$ may be written as

$$
g(F(x)) = g_{t(F(x))}(F(x)) = g_{t(F_{s(x)}(x))}(F_{s(x)}(x)).
$$

For each $i = 1 \ldots, m$ and $j = 1, \ldots, l$, consider the set

$$
U_{ij} = \{x \in \mathbb{R}^p, \, s(x) = i, t(F_i(x)) = j\}.
$$

Fix $(i, j)$ in $\{1, \ldots, m\} \times \{1, \ldots, l\}$, by Lemma 2, $t \circ F_i$ is an elementary index on $\mathbb{R}^{p_1}$. Hence $U_{ij}$ is the solution set of finitely many equalities and inequalities involving functions in $\mathcal{E}$. We associate to the bi-index $(i, j)$ the corresponding set $U_{ij}$ and the function $g_j(F_i(x)) \in \mathcal{E}$. Note that we assumed that the composition is well defined. Identifying each pair $(i, j)$ with a number in $\{1, \ldots, nm\}$, we obtain an elementary selection for $g \circ F$ and hence $g \circ F \in \mathcal{S}$. $\square$

**Proof of Proposition 4:** The derivation formula follows from the proof argument of Proposition 1, for each pair $(i, j)$, the function $g_j \circ F_i$ is the composition of two $C^1$ functions and its gradient is given by $J_{F_i} \times \nabla g_j \circ F_i$ on $U_{ij}$. By construction of $U_{ij}$ and definition of the selection derivative, this corresponds to (5) on $U_{ij}$ and the result follows. $\square$

**Proof of Lemma 1:** We actually prove a slightly stronger result, namely for each $i \in \{p+1, \ldots, m-1\}$

$$P_i \left(I - e_{i+1}e_{i+1}^T + d_{i+1}e_{i+1}^T\right) \ldots \left(I - e_m e_m^T + d_m e_m^T\right) = P_i \left(I + d_{i+1}e_{i+1}^T\right) \ldots \left(I + d_m e_m^T\right) \tag{9}$$

We argue by exhaustion from $i = m-1$ downward to $i = p$, which is the result of interest. If $i = m-1$, we indeed have

$$P_{m-1}\left(I - e_m e_m^T + d_m e_m^T\right) = P_{m-1}\left(I + d_m e_m^T\right)$$

since $P_{m-1} e_m e_m^T = 0$. Now assume that (9) holds true for an index $i$ within $\{p+1, \ldots, m-1\}$, then we have

$$
\begin{aligned}
& P_{i-1}\left(I - e_i e_i^T + d_i e_i^T\right) \ldots \left(I - e_m e_m^T + d_m e_m^T\right) \\
=~& P_{i-1}\left(I - e_i e_i^T + d_i e_i^T\right)\left(I - e_{i+1}e_{i+1}^T + d_{i+1}e_{i+1}^T\right) \ldots \left(I - e_m e_m^T + d_m e_m^T\right) \\
=~& P_{i-1}\left(I - e_i e_i^T + d_i e_i^T\right) P_i \left(I - e_{i+1}e_{i+1}^T + d_{i+1}e_{i+1}^T\right) \ldots \left(I - e_m e_m^T + d_m e_m^T\right) \\
=~& P_{i-1}\left(I + d_i e_i^T\right) P_i \left(I + d_{i+1}e_{i+1}^T\right) \ldots \left(I + d_m e_m^T\right) \\
=~& P_{i-1}\left(I + d_i e_i^T\right)\left(I + d_{i+1}e_{i+1}^T\right) \ldots \left(I + d_m e_m^T\right),
\end{aligned}
$$

where step 1 is expanding the product, step 2 is because $P_{i-1}P_i = P_{i-1}$ and $e_i^T P_i = e_i^T$, step 3 combines the fact that $P_{i-1}e_i = 0$ and (9) which we assumed to be true, the last step uses again the fact that $P_{i-1}P_i = P_{i-1}$ and $e_i^T P_i = e_i^T$. Hence the result holds by exhaustion. $\quad\square$

**Proof of Proposition 6:** Consider the sequence $s_k = S(x_k)$, by taking a subsequence we may assume that $s_k$ is constant, say equal to $\{1, \ldots, r\}$. Hence for all $k$, $v_k \in \operatorname{conv}\left(\{\nabla f_i(x_k), i = 1, \ldots r\}\right)$ and $f(x_k) = f_i(x_k), i = 1, \ldots, r$. Passing to the limit, we have $f(\bar{x}) = f_i(\bar{x}), i = 1, \ldots, r$ and hence $\{1, \ldots, r\} \in S(x)$. Furthermore, $\bar{v} \in \operatorname{conv}\left(\{\nabla f_i(\bar{x}), i = 1, \ldots r\}\right) \subset D_f(\bar{x})$. $\quad\square$

# C  o-minimal structures, definability and conservative fields

## C.1  $(\mathbb{R}, \exp)$-definability

We recall here the results of geometry that we use in the present work. Some references on this topic are [19, 21].

An *o-minimal structure* on $(\mathbb{R}, +, \cdot)$ is a collection of sets $\mathcal{O} = (\mathcal{O}_p)_{p \in \mathbb{N}}$ where each $\mathcal{O}_p$ is itself a family of subsets of $\mathbb{R}^p$, such that for each $p \in \mathbb{N}$:

   (i) $\mathcal{O}_p$ is stable by complementation, finite union, finite intersection and contains $\mathbb{R}^p$.

   (ii) if $A$ belongs to $\mathcal{O}_p$, then both $A \times \mathbb{R}$ and $\mathbb{R} \times A$ belong to $\mathcal{O}_{p+1}$;

   (iii) if $\pi : \mathbb{R}^{p+1} \to \mathbb{R}^p$ is the canonical projection onto $\mathbb{R}^p$ then, for any $A \in \mathcal{O}_{p+1}$, the set $\pi(A)$ belongs to $\mathcal{O}_p$;

   (iv) $\mathcal{O}_p$ contains the family of real algebraic subsets of $\mathbb{R}^p$, that is, every set of the form

$$\{x \in \mathbb{R}^p \mid g(x) = 0\}$$

   where $g : \mathbb{R}^p \to \mathbb{R}$ is a polynomial function;

   (v) the elements of $\mathcal{O}_1$ are exactly the finite unions of intervals.

A subset of $\mathbb{R}^p$ which belongs to an o-minimal structure $\mathcal{O}$ is said to be *definable in $\mathcal{O}$*. A function is *definable in $\mathcal{O}$* whenever its graph is definable in $\mathcal{O}$). A set valued mapping (or a function) is said to be definable in $\mathcal{O}$ whenever its graph is definable in $\mathcal{O}$. The terminology *tame* refers to definability in an o-minimal structure without specifying which structure.

The simplest o-minimal structure is given by the class of real semialgebraic objects. Recall that a set $A \subset \mathbb{R}^p$ is called *semialgebraic* if it is a finite union of sets of the form

$$\bigcap_{i=1}^k \{x \in \mathbb{R}^p \mid g_i(x) < 0,~ h_i(x) = 0\}$$

where the functions $g_i, h_i : \mathbb{R}^p \to \mathbb{R}$ are real polynomial functions and $k \geqslant 1$. The key tool to show that these sets form an o-minimal structure is Tarski-Seidenberg principle which ensures that (iii) holds true.

According to [42], there is an o-minimal structure which contains all semialgebraic sets and the graph of the exponential function, we fix this o-minimal structure and call it $\mathcal{O}$. As a consequence, all functions which can be described by a finite compositional expression involving polynomials, quotients, exponential and logarithms are definable in $\mathcal{O}$. In particular any function $f \in \mathcal{S}$ is definable in $\mathcal{O}$, which opens the use of powerful geometric tools [19, 21] for functions in $\mathcal{S}$. From now on, we call an object *definable* if it is definable in $\mathcal{O}$.

As detailed in [19] the following holds true

**Proposition 7 (Quantifier elimination)** *Any first order formula (quantification on variables only) involving definable functions and definable sets describes a definable set.*

This allows to prove Claim 1

**Proof of Claim 1:** The function $t \mapsto s(x + t(y - x))$ is definable and has values in $\{1, \ldots, m\}$. For each $j \in \{1, \ldots, m\}$, the set $S_j = \{t \in [0, 1], \ s(x + t(y - x)) = j\}$ is definable, and by (v), it is a finite union of intervals. For each $j$ consider only the endpoints of those intervals with nonempty interior, this provides the desired partition. $\qquad\square$

### C.2 Properties of definable sets

The tangent space at a point $x$ of a manifold $M$ is denoted by $T_x M$. Given a submanifold[6] $M$ of a finite dimensional Riemannian manifold, it is endowed by the Riemanninan structure inherited from the ambient space. Given $f : \mathbb{R}^p \to \mathbb{R}$ and $M \subset \mathbb{R}^p$ a differentiable submanifold on which $f$ is differentiable, we denote by $\mathrm{grad}_M f$ its Riemannian gradient or even, when no confusion is possible, $\mathrm{grad}\, f$.

A $C^r$ stratification of a (sub)manifold $M$ (of $\mathbb{R}^p$) is a partition $\mathcal{S} = (M_1, \ldots, M_m)$ of $M$ into $C^r$ manifolds having the property that $\mathrm{cl}\, M_i \cap M_j \neq \varnothing$ implies that $M_j$ is entirely contained in the boundary of $M_i$ whenever $i \neq j$. Assume that a function $f : M \to \mathbb{R}$ is given and that $M$ is stratified into manifolds on which $f$ is differentiable. For $x$ in $M$, we denote by $M_x$ the strata containing $x$ and we simply write $\mathrm{grad}\, f(x)$ for the gradient of $f$ with respect to $M_x$.

Stratifications can have many properties, we refer to [21] and references therein for an account on this question and in particular for more on the idea of a Whitney stratification that we will use repeatedly. We pertain here to one basic definition: a $C^r$-stratification $\mathcal{S} = (M_i)_{i \in I}$ of a manifold $M$ has the *Whitney-(a) property,* if for each $x \in \mathrm{cl}\, M_i \cap M_j$ (with $i \neq j$) and for each sequence $(x_k)_{k \in \mathbb{N}} \subset M_i$ we have:

$$\left. \begin{array}{l} \lim_{k \to \infty} x_k = x \\[2mm] \lim_{k \to \infty} T_{x_k} M_i = \mathcal{T} \end{array} \right\} \implies T_x M_j \subset \mathcal{T}$$

where the second limit is to be understood in the Grassmanian, i.e., "directional", sense. In the sequel we shall use the term *Whitney stratification* to refer to a $C^1$-stratification with the Whitney-(a) property. The following can be found for example in [21].

**Theorem 5 (Whitney stratification)** *Let $A_1, \ldots, A_k$ be definable subsets of $\mathbb{R}^p$, then there exists a definable Whitney stratification $(M_i)_{i \in I}$ compatible with $A_1, \ldots, A_k$, i.e. such that for each $i \in I$, there is $t \in \{1, \ldots k\}$, such that $M_i \subset A_t$.*

This allows for example to prove Claim 2

**Proof of Claim 2:** The sets $V_1, \ldots, V_m$ form a definable partition of $\mathbb{R}^p$. Consider a Whitney stratification of $\mathbb{R}^p$, $(M_i)_{i \in I}$ compatible with the closure of $V_1, \ldots, V_m$. The boundary of each $V_i$ is a finite union of strata of dimension strictly smaller than $p$ and hence has measure zero. The remaining strata (open of maximal dimension) have to be dense in $\mathbb{R}^p$ since we started with a partition. $\qquad\square$

## C.3 Variational stratification and projection formulas

**Definition 6 (Variational stratification)** Let $f\colon \mathbb{R}^p \to \mathbb{R}$, be locally Lipschitz continuous, let $D\colon \mathbb{R}^p \rightrightarrows \mathbb{R}^p$ be a set valued map and let $r \geqslant 1$. We say that the couple $(f, D)$ has a $C^r$ *variational stratification* if there exists a $C^r$ Whitney stratification $\mathcal{S} = (M_i)_{i \in I}$ of $\mathbb{R}^p$, such that $f$ is $C^r$ on each stratum and for all $x \in \mathbb{R}^p$,

$$\mathrm{Proj}_{T_{M_x}(x)} D(x) = \{\mathrm{grad}\, f(x)\}, \tag{10}$$

where $\mathrm{grad}\, f(x)$ is the gradient of $f$ restricted to the active strata $M_x$ containing $x$.

The equations (10) are called *projection formulas* and are motivated by Corollary 9 in [11] which states that Clarke subgradients of definable functions have projection formulas.

Let us recall the definition of conservative set-valued mappings from [12] and one of its characterization.

**Definition 7 (Conservative set-valued mappings)** Let $f$ be a Lipschitz continuous function. A set valued vector field $D$ is called *conservative* if for any absolutely continuous path $\gamma\colon [0, 1] \mapsto \mathbb{R}$, we have

$$f(\gamma(1)) - f(\gamma(0)) = \int_0^1 \min_{v \in D(\gamma(t))} \langle v, \dot{\gamma}(t) \rangle \, dt = \int_0^1 \max_{v \in D(\gamma(t))} \langle v, \dot{\gamma}(t) \rangle \, dt. \tag{11}$$

Equivalently $D$ is conservative for $f$, if for all absolutely continuous curves $\gamma\colon [0, 1] \mapsto \mathbb{R}^p$, for almost all $t \in [0, 1]$, $f \circ \gamma$ is differentiable and

$$\frac{d}{dt} f(\gamma(t)) = \langle v, \dot{\gamma}(t) \rangle, \qquad \forall v \in D(\gamma(t)).$$

The following combines other results from [12], where one implication is essentially due to [20] based on [11].

**Theorem 6 (Characterization of conservativity)** *Let $D\colon \mathbb{R}^p \rightrightarrows \mathbb{R}^p$ be a definable, nonempty compact valued, graph closed set valued field and $f\colon \mathbb{R}^p \mapsto \mathbb{R}$ be a definable locally Lipschitz function. Then the following are equivalent*

- *$D$ is conservative for $f$.*
- *For any $r \geqslant 1$, $(f, D)$ admit a $C^r$ variational stratification.*

This result allows to prove the following

**Proof of Theorem 3:** We prove that there is a $C^1$ projection formula (see Theorem 6). For each $I \subset \{1, \ldots, m\}$, set $V_I = \{x \in \mathbb{R}^p, S(x) = I\}$. On each set $V_I$, $f(x) = f_i(x)$ for all $i \in I$. These sets are definable, hence, there is a definable Whitney stratification of $\mathbb{R}^p$ which is compatible with them (Theorem 5). For any $C^1$ manifold $M$ in the stratification there is an index set $I \subset \{1, \ldots, m\}$ such that for all $i \in I$ and all $x \in M$, $f(x) = f_i(x)$ and $S(x) = I$. Since each $f_i$, $i \in I$ is $C^1$ and they agree on $M$, they represent the same function when restricted to $M$. Hence they have the same differential on $M$ and since they are all globally $C^1$ this agrees with the projection of their gradient on the tangent space of $M$. Hence the projection of $D_f(x)$ to the tangent space to $M$ at $x$ is single valued and corresponds to the derivative of $f$ restricted to $M$. This is sufficient to conclude as this is precisely the variational stratification required by Theorem 6. □

# D Convergence to selection critical points

**Proof of Theorem 4, first part:** We use here the results on conservative fields developed in [12]. To prove the theorem it suffices to establish that:

- $D_J$ is a conservative field for $J$
- the number of $D_J$ critical values are finite.

The first point is Theorem 6 while the second one is the consequence of the latter and the definability of the couple $f, D_f$, see Proposition 8 (ii). To conclude it suffices to apply the convergence results in [12, Theorem 9]. $\square$

**Proof of Theorem 4, second part:** This result is a consequence of the more general Theorem 7 established in Section E. Let $F$ be the finite set given in Theorem 7, the set

$$\{c \in (0,1], \exists k \in \mathbb{N}, c\gamma_k \in F\},$$

is countable, and hence has zero measure. So for almost all $c \in (0,1]$, $\{c\gamma_k\}_{k \in \mathbb{N}}$ does not intersect $F$. Using Theorem 7, there is a zero measure set $N$ such that any initialization outside $N$ provides almost surely a subgradient sequence. By hypothesis, for almost every $x_0 \in K \backslash N$, the sequence is bounded almost surely and the result follows from Theorem 7. $\square$

# E Artificial critical points

Being given a Lipschitz continuous function on $\mathbb{R}^p$ and a conservative field $D$, one has two types of $D$-critical points:

- Clarke critical points: $\partial^c f(x) \ni 0$, we denote the set of these points by $\text{crit}^c f$
- Artificial critical points $\partial^c f(x) \not\ni 0$ and $D(x) \ni 0$, we denote this set by $\text{crit}^a f$

Critical values are defined accordingly as images of critical points.

**Proposition 8 (Artificial critical points)** *Assume $f : \mathbb{R}^p \to \mathbb{R}$ and $D : \mathbb{R}^p \rightrightarrows \mathbb{R}^p$ are definable in a common o-minimal structure. The connected components $C_i$ of $\text{crit}^a f$, which are in finite number, satisfy*

*(i) $\dim C_i < p$*

*(ii) $f(C_i)$ is a singleton, and as a consequence the $D$ critical values of $f$ are in finite number,*

*(iii) $\text{crit}^a f$ does not contain local minimum (nor local maximum)*

**Proof :** By definability of $\text{crit}^a f$, the number of connected components is finite.

If $C_i$ had full dimension it would contain a non trivial ball on which $f$ should be constant by the integral property. This would in turn imply that the points in the ball would also be local minimum and thus Clarke critical, which is impossible.

To see that the critical values are in finite number it suffices to evoke the fact that Clarke critical values are finite [11] and use that artificial critical values are in finite number.

By definability the connected components are arcwise-connected with piecewise $C^1$ paths. Using the integral property this shows $f$ is constant on $C_i$.

(iii) is obvious since local minimum or maximum are Clarke critical. $\square$

As explained in the introduction, artificial critical points are "computing artefacts", whence their names. For algorithmic differentiation the "gradient" provided by a program is zero while the point might even be a smooth non critical point. We consider the setting of the mini-batch algorithm of the last section.

**Theorem 7** *Assume that each $f_1, \ldots, f_n$ belongs to $\mathcal{S}$. There exists a finite subset of steps $F \subset (0, +\infty)$ and a zero measure meager subset $N$ of $\mathbb{R}^p$, such that for any positive sequence $\gamma_k = o(1/\log k)$ avoiding values in $F$, and any almost surely bounded sequence with initial condition in $\mathbb{R}^p \backslash N$, we have*

- *$J(x^k)$ converges towards a Clarke critical value almost surely,*

- *the cluster points of $x^k$ are Clarke critical point almost surely.*

**Proof :** The proof is twofold. We first prove that the set of initial conditions leading to an artificial critical point or more generally to a non differentiability point within a finite time is "small". We then use this fact to conclude.

**Claim 3** *Let $g : \mathbb{R}^p \to \mathbb{R}$ be a definable differentiable function. Set, for $\lambda > 0$,*

$$\Phi_\lambda = \lambda Id - \nabla g,$$

*where $Id$ denotes the identity. There exists a finite set $F$ in $(0, +\infty)$ such that,*

$$\forall \lambda \in (0, +\infty) \backslash F, \ \forall Z \subset \mathbb{R}^p \text{ definable }, \dim Z < p \Rightarrow \dim \Phi_\lambda^{-1}(Z) < p. \qquad (12)$$

Proof of the claim. Denote by $L$ the set of points where $g$ is twice differentiable so that $L$ is dense and definable. Denote by $\lambda_1, \ldots, \lambda_p : L \to \mathbb{R}$ a representation of the eigenvalues of $\nabla^2 g$. Refine $L$ to be contained in the common domain of differentiability for each $\lambda_i$, $L$ remains open and dense. By the definable Sard's theorem the critical values of each function $\lambda_i$ is finite, so that the set of all these values which we denote by $F$ is itself finite.

Take a positive real $\lambda \notin F$ and consider the set

$$K_\lambda := \{x \in L : \Phi_\lambda'(x) = \lambda Id - \nabla^2 g(x) \text{ is not invertible}\}.$$

By diagonalization, we see that the determinant of $\Phi_\lambda'(x)$ is $\prod_{i=1}^{d}(\lambda - \lambda_i(x))$ for any $x$, thence

$$K_\lambda \subset \bigcup_{i=1}^{m} \{x \in L, \ \lambda_i(x) = \lambda\}.$$

Since $\lambda$ is a regular value for each $\lambda_i$ the previous set is a finite union of manifolds of dimension $p-1$, see e.g., [19]. This implies that the set $\mathbb{R}^p \backslash K_\lambda = \{x \in L : \Phi_\lambda'(x) \text{ is invertible }\}$ is dense. Using the above, we deduce that there exists finitely many open connected subsets $U_1, \ldots, U_r \subset L$ of $\mathbb{R}^p \backslash K_\lambda$ such that $U_1 \cup \ldots \cup U_r$ is dense in $L$ and thus in $\mathbb{R}^p$. Take now $Z \subset \mathbb{R}^p$ definable with $\dim Z < p$. Assume towards a contradiction that there exists a nonempty open ball $B$ in $\Phi_\lambda^{-1}(Z)$. In that case $B$ must have a nonempty intersection with some $U_{i_0}$. The set $\Phi_\lambda(B \cap U_{i_0})$ is open because $\Phi_\lambda$ is a diffeomorphism on $U_i$ on its image. Since on the other hand we have $\Phi_\lambda(B \cap U_{i_0}) \subset Z$, we have a contradiction and the claim is proved. $\qquad\square$

For each $I \subset \{1, \ldots, n\}$, we denote by $f_{I,1}, \ldots, f_{I,m_I}$ the bricks attached to $f_I$ where $m_I \geqslant 1$. Denote by $\mathrm{Sing}$ the set of points on which at least one $f_I$ is non differentiable and $C$ the set of points for which $\widehat{\nabla} f_I \neq \nabla f_I$ for at least one $I$. By Proposition 3 and definability, $\mathrm{Sing}$ and $C$ are finite unions of manifolds of dimension at most $p-1$.

Set $\Phi_{I,j}^k = Id - \gamma_k \nabla f_{I,j}$, with $I \subset \{1, \ldots, m\}$, $j \in \{1, \ldots, m_I\}$ and $Id$ denotes the identity. Applying Claim 3, we can find a finite set $F$ for which $\gamma_k \notin F$ implies that each $\Phi_{I,j}^k$ has the property (12). Indeed, for each $I \subset \{1, \ldots, m\}$, $j \in \{1, \ldots, m_I\}$, there is $F_{I,j} \subset \mathbb{R}$ finite such that $f_{I,j}$ has property (12). Since the subsets $I$ are in finite number and each $m_I$ is finite, the set $F = \bigcup_{I \subset \{1,\ldots,m\}} \bigcup_{j \in \{1,\ldots,m_I\}} F_{I,j}$, is also finite. For each $k \in \mathbb{N}$, $I \subset \{1, \ldots, m\}$, $j \in \{1, \ldots, m_I\}$. Remark that if $\gamma_k \notin F$ then $\Phi_{I,j}^k$ has property (12).

For $k \leqslant k_0$ fixed, let us consider the finite set of definable mappings defined by

$$\Psi_{k_0} := \left\{ \prod_{j=1}^{k} \Phi_{I_j, i_j}^j : k \leqslant k_0, I_j \subset \{1, \ldots, n\}, i_j \in \{1, \ldots, m_{I_j}\} \right\}.$$

We now assume that $\gamma_k \notin F, \forall k \geqslant 0$, so that each mapping in $\Psi_{k_0}$ has the property (12) and

$$N_{k_0} := \{x \in \mathbb{R}^p : \exists k \leqslant k_0, \exists \Phi \in \Psi_k(x) \in C \cup \mathrm{Sing}\}$$

These are initial conditions in $U$ leading to an artificial critical or a non-differentiability point within $U$ before time $k_0$.

We can also write

$$N_{k_0} \subset \bigcup_{\Phi \in \Psi_{k_0}} \Phi^{-1}\left(C \cup \mathrm{Sing}\right).$$

From stratification arguments we know that $\mathrm{Sing}$ has a dimension lower than $p-1$. On the other hand, $C$ has dimension strictly lower than $p$ by Proposition 3. Claim 3 applies and yields

$\dim \Phi^{-1}(C \cup \mathrm{Sing}) < p$ for all $\Phi \in \Phi_{k_0}$. As a consequence $N_{k_0}$ is closed with nonempty interior and so does $N := \cup_{k \in \mathbb{N}} N_k$ by Baire's theorem. Similarly $N$ has zero measure as a countable union of zero measure sets.

This proves that any sequence with initial condition out of $N$ must remain in the zone of differentiability of $J$ as well as all $f_I$. In particular if $I$ is taken uniformly at random among possible subsets, for all $x \notin N$, we have $\mathbb{E}_I[\widehat{\nabla} f_I(x)] = \widehat{\nabla} J(x) = \nabla J(x) = \partial^c J(x)$, so that these specific sequences can also be seen as stochastic subgradient sequences for $J$. To be more specific, the sequence $x_k$ can be seen as one of the sequence generated by the algorithm

$$y_{k+1} \in y_k - \gamma_k \partial^c J(y_k) + \epsilon_k$$

where $\epsilon_k$ is a random noise with zero mean. Using general results [20, 5], we know that $y_k$ sequences, when bounded almost surely, have limit points which are Clarke critical. $\square$

## Footnotes

[6]We only consider embedded submanifolds