[Reviews · NeurIPS 2020]

Review 1

Summary and Contributions: The paper's starting point is that automatic differentiation (AD) acts on programs, not functions, and may either not return the expected results (namely the left or right-derivatives of a function), or return inconsistent results, notably when two different programs represent the same function. To study the behavior of AD, the authors introduce elementary selection, on which AD acts, and selection derivatives which AD returns. Properties of selection derivatives are derived, such as integration along segments and a chain rule. More generally, the article introduces a calculus of elementary selection. The paper ends by analyzing the asymptotic properties of an optimizer, for which the gradient are computed using AD, thereby illustrating the benefit of the presented framework.

Strengths: The paper addresses an important issue with AD, when applied to non-smooth functions. Even when the derivative is ill-defined, AD persists in returning an output. It's good to emphasize this and develop a formal framework to describe what AD actually does. The paper succeeds in motivating the research and clearly defines a framework for studying the problem at hand. Pedagogically very sound, and a nice discussion of a underappreciated topic. The lemmas and theorems are non-trivial and suggests future extensions of interest. The final section illustrates how to incorporate the properties of AD in the study of algorithms, which suggests important applications.

Weaknesses: The paper doesn't discuss much the implications of the here developed theory, which makes it difficult to evaluate the significance and relevance to the Neurips community. Where do we next develop the theory? How should the gained insights impact AD libraries such as TensorFlow? In which context are selection derivatives desirable or undesirable? It may be difficult to address these additional questions, given the constraining format, but the absence of conclusion (and broader impact statement) strikes me as a missed opportunity.

Correctness: The initial examples, which constitute the only empirical component, look good. I would detail the underlying program for the relu function, which acts as an IF statement, conditioning on x <= 0 (-- I'm guessing based on Figure 1). If the condition is x < 0, the output of AD would be different. You could add the corresponding expression graphs (optional, in the supplement...), which would nicely preface elementary selection. I haven't checked the proofs in the supplementary material. There were several typos, notably indices, in the statements of lemma and theorems; for example, proposition 4, when describing the domain of J_f. These need to be corrected. Also, please define the notation. For example, the term "conv" should be defined upfront, as it may have different meanings for Neurips' diverse audience.

Clarity: Yes. The introduction is very clear and does a great job setting up the problem. I would emphasize that at 0, relu is not differentiable -- and raise the question of what would we expect AD to return. The authors opt for 0, but this choice should be briefly justified. There are indeed cases where it is better to flag non-smooth functions and break, rather than assign some value to their "numerical derivative". Once I reached section 5, I wasn't sure where the paper was going. The absence of a conclusion made the work feel unfinished. A summary of the main points and future goals would improve the paper. There were several typos. Please proofread carefully.

Relation to Prior Work: The paper points to several references and discusses how their work differs in section 1.2. But the claims can only be verified by reading the references. A more detailed discussion of the existing references (optional, in the supplement...) could be helpful. Neither the (very general) title, nor the abstract convey how the here presented work departs from the existing literature.

Reproducibility: Yes

Additional Feedback: Suggestions for the broader conclusion / impact statement: (i) the paper raises awareness about an important issue for a ubiquitous tool, and (ii) it invites us to reason about the differentiation mechanism when studying the property of algorithms, which constitutes, in my view, an interesting avenue for future research. ===== After rebuttal ====== I look forward to the updates the authors propose to make. Notably the discussion on the implication of the presented work (also requested by R2) and the addition of a conclusion. I keep my score unchanged.


Review 2

Summary and Contributions: the authors highlight that automatic differentiation of non-smooth functions depends on program, rather than being fully determined by the function that program implements. they introduce a elementary selection functions to describe non-smooth functions, and a convex automatic differentiation method defined over these functions. they use this class of functions to define a convex automatic differentiation operator and to demonstrate that the spurious behavior at critical points does not affect asymptotic behavior under stochastic gradient methods. contributions: theoretical: define a broad class of non-smooth nonconvex functions over which to describe or define automatic differentiation theoretical: define selection derivatives and generalize calculus rules that cope with spurious behaviors at boundaries/critical points between selections. theoretical: introduce convex differentiation operator given selection functions theoretical: use framework to show that all accumulation points of the algorithm are clarke critical for randomly initialized minibatch SGD

Strengths: strong theoretical grounding. addresses relevant questions about the stability of automatic differentiation, and interesting conceptual questions about a widespread interpretation of automatic differentiation as providing derivatives of the functions, which neglect the programmatic nature of automatic differentiation. proposes a formal treatment of nonsmooth functions that permits analysis of the problems

Weaknesses: in a few places, the authors claim that these artificial critical points pose a problem in practice for training models and that false outputs are met in practice (eg line 29-30, 77-78, 225-226). however, they do not provide evidence for this claim: there is no citation, proof, or experimental demonstration. their last result also seems to contradict this (at least in the case of stochastic gradient methods), where they show that artificial critical points does not affect asymptotics for random initializations. this is a substantial concern since the implications and contributions of this paper depend substantially on this point: it's not entirely clear whether their contribution is to (1) introduce a theoretical framework that helps us address the real world problem of artificial critical points, or (2) whether the contribution is to introduce a theoretical framework that, among other things, allows us to explain why these are not an issue. it was not clear how different selection functions within a neural network would work together -- it seems that if units with different selection functions were combined in a single architecture, the number of selections would grow very rapidly and that the selection derivatives would scale poorly. would be helpful if the authors clarified how to think about this in the deep learning context given their intro and results discuss deep learning. ---- after rebuttal ---- thanks to the authors for their clarifications. their comments mitigate my concerns, however i still would suggest rewriting to make the theoretical contributions clearer for a broader neurips audience.

Correctness: as far as i know yes.

Clarity: the first 4 sections were very clearly written, and i followed the paper pretty well until section 5. however, section 5 was difficult to follow. for instance, in definition 5, terms are not defined (double arrow? conv?). this might be standard notation in another field but was not familiar to me. theorem 3 requires a lot of reference to equations that need to be hunted down. it was also not clear to me in section 5 the exact role of the selection functions. were they using them to describe existing automatic differentiation implementations, and therefore showing that artefacts are not significant with SGD? or were they introducing a new automatic differentiation, which solved the issue of artefacts? this point bears substantially on the interpretation of the contributions of the paper. Fig 2 was also bit hard to follow, and i think the authors could omit it without hurting their main points.

Relation to Prior Work: as far as i can tell, yes.

Reproducibility: Yes

Additional Feedback:


Review 3

Summary and Contributions: The authors present a general framework to study derivative computed by automatic differentiation for non-smooth functions. They introduce selection derivatives, provide simple calculus rules for them and present how automatic differentiation generically computers selection derivatives. They show that selection derivatives are conservative fields and use this result to demosntrate almost sure convergence to stationary points by stochastic gradient descent.

Strengths: The general presentation of the paper is outstanding, the authors provide a very clear introduction to the concepts they present and to automatic differentiation for non-smooth functions with illustrative examples. Contrary to previous works, the framework presented by the authors does not constraint the set of programs on which automatic differentiation operates. It rather focuses on the properties of the derivatives that automatic differentiation provide to get convergence guarantees of the same flavor as previous work. Overall the framework is a sound basis for the analysis of wilder usages of automatic differentiation. Again, the quality of the presentation itself makes it a very valuable work for the community.

Weaknesses: The bounds are asymptotic and no convergence rate is proven. Note that there exists no work up to my knowledge that gives convergence rates in this case, so I point it out as a weakness but I believe the clarity and the generality of the framework already have a strong value for the community.

Correctness: I have not checked in details theorem 4. Other results are correct for me.

Clarity: As said above, the clarity of the paper is outstanding.

Relation to Prior Work: The relation to relevant previous work is clearly detailed. In particular, it provides a clear stream of contributions not restricted to recent ones.

Reproducibility: Yes

Additional Feedback: The present framework assumes continuity of the functions represented by the programs (definition 3). Some deep networks use argmax mapings (typically the argmax of a scalar product over a polytope, see e.g. SparseMAP: Differentiable Sparse Structured Inference, Niculae et al). These argmax functions can be defined as elementary index, however their composition with other elementary functions produce piecewise continuous and not continuous functions. It seems to me that the present framework may apply in this case but I may have misunderstood some details. A detailed discussion may be of interest to potentially enlarge the applications of the framework. Minor details: line 103: Remark 1 does not refer to anything line 112: space between an and elementary line 130: for x > 0 Prop 4: q -> p2, p -> p1 Proof of Thm1: Shouldn't it be A(P_{r,s]) \ni A(f)(s) + r since the assumption is that 0 \in \partial^A(rely) at 0? Claim 1 : add "is constant on [a_i, a_{i+1}]" o-minimal def: shouldn't it be the set of positive integers and not the set of integers? ===== After rebuttal ====== I thank the authors for their answer and for the overall presentation of the paper. I keep my score unchanged.

[Author Response · NeurIPS 2020]

We thank the referees for their comments.

Following the suggestions of R1 and R2, we will add a conclusion to the paper and modify the broader impact statement.
This will clarify our contribution, its practical impact, but also the results described in Section 5. We will, of course,
implement the other suggestions made by the reviewers, such as adding mathematical definitions, correcting typos, and
proofreading.

**Response to R1:**   We will indeed add a more detailed discussion on existing work in the supplementary material,
this is a good idea. We will also provide an alternative graphical representation of the relu function using a different
selection.

**Response to R2:**   Our present contribution is to provide a simple theoretical framework, which allows us to describe
current implementations of AD. We also explain why, *in theory*, artificial critical points are not an issue (item (2)
suggested by R2). Claiming that spurious critical points are met in the practice of deep learning indeed needs
developments that we have not provided (and that we cannot present in this paper). Instead, we shall illustrate their
existence through toy models; we will avoid saying that the problem can be met in practice. Now let us discuss the
main issue raised by R2: does the spurious behavior impact *practice*?

• Our "trap-avoidance" result suggests that, at high precision, artificial critical points are not met in practice.

• On the other hand, at low precision, "genericity results" may partly collapse. For instance, very large scale
linear programs have several solutions, even in 32 bits. Yet the theory says uniqueness is generic. It raises the
question: what is the impact of the spurious behavior at low precision?

These aspects will be briefly evoked, but they will be treated in a seperate work.

Line 29-30 is not a claim about the practical impact of artificial critical points. It is rather a statement about how
difficult the study of the *"spurious set"* can be when dimension grows. Indeed, in larger dimensions, the composition of
intermediate functions may be extremely complex, making the geometry of artificial point difficult to grasp. We will
rephrase so that this appears more clearly.

Regarding the combinatorial nature of selection derivatives, the reviewer is right, combining different selection functions
within a neural architecture leads to a combinatorial explosion of the number of possible selections, depending on the
choice for each unit. But the user does not meet this issue because

• The selection process is implicitly described by the numerical code describing the network (Proposition 5)! In
other words, practitioners already implement selection functions and selection derivatives by writing programs.
Selections are mathematical objects representing programs. They are only used for proofs and theory; for
implementation, one uses the corresponding programs!

• A given function has an infinity of selection representations, each inducing a potentially different selection
derivative. All our results work for *any such selection derivative*. So one does not need to consider all possible
selections, being able to compute a single of them is sufficient.

In other words, the scaling issue *has no impact whatsoever* and AD can be used, as is, to compute a selection derivative,
which is sufficient for optimization purposes.

On the other hand, the reviewer raises a very interesting question regarding the numerical behavior of AD related to
combinatorial explosion. A detailed presentation would require another complete paper, and we decided to postpone
this question for future research.

**Response to R3:**   The reviewer is absolutely right to point out rates and complexity. Contrary to the smooth setting,
we are not even sure what is the correct notion of approximate stationarity that should be used. This is a relevant topic
of future research, and we will add it to the (new) conclusion.

Regarding selection derivatives for discontinuous functions, the proposed framework can indeed be used. Actually,
Tensorflow provides a value for the derivative of the step function (zero everywhere) so selection derivatives for
discontinuous functions are already available in a sense. We believe that dropping the continuity constraint, one could
prove similar results as propositions 1,3, and 4. Hence this seems innocuous from a computational perspective.

The issue with discontinuous functions is that the current "theory" does not apply directly. It is much more difficult
to make a theoretically consistent interpretation of selection derivatives. In particular, Proposition 2 does not hold
anymore; it is not possible to integrate along segments. This property is actually crucial to prove convergence to critical
points. We plan to investigate these questions in future work.

[Meta-Review · NeurIPS 2020]

The paper discusses the widely recognized phenomenon that automatic differentiation of mathematical functions represented as code may produce inconsistencies. The paper makes concrete contributions beyond the existing literature on e.g. Clarke gradients, by defining a restricted but large class of functions on which the correctness of the proposed AD can be confirmed. R2 raises valuable questions about the implications of the results in practice, but (a) the rebuttal gracefully accedes these points, and makes a convincing promise to "avoid saying that the problem can be met in practice", and (b) I believe that the points made in the paper are worth addressing even if in practice they may have less alarming implications.